# Can a nation-wide e-cohort of ADHD and ASD in childhood be established using Welsh routinely available datasets?

Kate Langley [iD],[1] Marcos Del Pozo-Banos,[2,3] Søren Daalsgard,[4,5,6] Shantini Paranjothy [iD],[7] Lucy Riglin,[3,8] Ann John [iD],[2,3] Anita Thapar [iD] [3,8]

For numbered affiliations see end of article.

**Correspondence to**
Dr Kate Langley;
LangleyK@cardiff.ac.uk

## ABSTRACT

**Objectives** We investigated the feasibility and validity of establishing a nationwide e-cohort of individuals with a diagnosis of attention deficit hyperactivity disorder (ADHD) and/or autism spectrum disorder (ASD) for future longitudinal research.

**Design** Individuals with a childhood diagnosis of ADHD/ASD as recorded on routinely available healthcare datasets were compared with matched controls and a sample of directly assessed individuals with ADHD.

**Setting** This study used data from the Welsh Secure Anonymised Information Linkage Databank in Wales, UK. Routinely collected data from primary care, emergency department and hospital admissions were linked at person level.

**Participants** All individuals in Wales, UK born between 1 January 1991 and 31 December 2000. Individuals with a recorded diagnosis of ADHD and/or ASD by age 18 years were identified using International Classification of Diseases, 10th Revision and National Health Service (NHS) READ codes and matched to 3 controls each and 154 individuals with ADHD recruited from an established research study.

**Outcome measures** Recorded service use for anxiety and depression, alcohol and drug use and self-harm including emergency department use in young adulthood (age 16–25 years).

**Results** 7726 individuals had a recorded diagnosis of ADHD (80% male) and 5001 of ASD (79% male); 1.4% and 0.9% of the population, respectively. Cox's regression analyses showed ADHD was associated with increased risks of anxiety/depression (HR: 2.36, 95% CI: 2.20 to 2.53), self-harm (HR: 5.70, 95% CI: 5.07 to 6.40), alcohol (HR: 3.95, 95% CI: 3.42 to 4.56), drug use (HR: 5.88, 95% CI: 5.08 to 6.80) and emergency department service use (HR: 1.36, 95% CI: 1.31 to 1.41). Those with ASD were at increased risk of anxiety/depression (HR: 2.11, 95% CI: 1.91 to 2.34), self-harm (HR: 2.93, 95% CI: 2.45 to 3.50) and drug use (HR: 2.21, 95% CI: 1.66 to 2.95) but not alcohol use. The ADHD e-cohort were similar to the directly assessed cohort.

**Conclusions** Our identification strategy demonstrated the feasibility of establishing a large e-cohort of those with ADHD/ASD with expected patterns of poorer early adult outcomes, demonstrating a valid method of identifying large samples for future longitudinal studies without selective attrition.

## STRENGTHS AND LIMITATIONS OF THIS STUDY

⇒ This study used population wide administrative data from a well-established and comprehensive databank.
⇒ Data from both primary and secondary care were combined.
⇒ The study used a directly assessed sample of those with attention deficit hyperactivity disorder to help validate the establishment of the administratively identified sample.
⇒ Not all individuals had passed through the whole follow-up period at the end of the study.
⇒ Studies using administrative data is dependent on data having been collected a recorded.

## INTRODUCTION

Attention deficit hyperactivity disorder (ADHD) and autism spectrum disorder (ASD) are neurodevelopmental conditions that typically onset in early childhood.[1] These conditions are known to be associated with multiple adverse outcomes.[2–4] One barrier to conducting longitudinal studies of neurodevelopmental and mental health disorders is selective attrition. Multiple studies have observed that individuals with mental health problems, with higher genetic loading for psychiatric disorders[5 6] and from socially disadvantaged backgrounds[7] are more likely to drop out of longitudinal research, potentially limiting generalisability of results, particularly to more severe cases. An alternative approach that has been used, is to take advantage of routinely collected healthcare data.

Wales is one of four UK nations with a population of around 3 million and has a number of devolved areas of administration, including health and social care. Wales hosts one of the most comprehensive anonymised record-linked data bases, SAIL (Secure Anonymised Information Linkage) Databank

(www.saildatabank.com) that includes health and social care data on every individual. These records include not only secondary care, emergency department contacts and prescription data but also uniquely encompass primary care and household level data. The aim of this study was to assess the feasibility of identifying those within Wales with a diagnosis of ADHD or ASD in childhood using record-linked routinely available healthcare data and investigate the validity of a nationwide e-cohort for future longitudinal clinical, health services, epidemiological and social sciences research. To achieve this aim, we first identified individuals with a childhood record of ADHD and/or ASD using the SAIL record-linked databases and compared them to matched controls for known early adult outcomes (aged 16–25) associated with ADHD and ASD. For those with ADHD, we also compared the e-cohort to a hybrid cohort of individuals with known ADHD diagnoses verified by direct interview.

## METHODS
### Study design and participants
The SAIL Databank is an electronic resource holding anonymised person-level linkable routinely available data from health and public settings for individuals in Wales, UK.[8] Each individual is assigned an anonymous linkage field which can then be used to identify their records across different datasets. This system is secure and anonymous, using a privacy-protecting split-file approach. In this study, we used the following datasets: the Welsh Demographic Service; the General Practice Database (GPD) (covering 77%, ie, 333/432, of all general practices (GPs) in Wales); the Emergency Department Dataset (EPD) for NHS Wales; and the Patient Episode Database for Wales (PEDW) (with hospital admissions data for both inpatient and day cases). Full details are given in online supplemental table S1.

### Individuals with ADHD or ASD in SAIL
We identified a list of codes from the International Classification of Diseases, 10th Revision (ICD-10) associated with ADHD and ASD from published literature and cross mapped them to READ codes, a coded thesaurus of clinical terms used in the UK National Health Service. We prepared an initial list of READ codes (diagnoses, medications and administrative codes) and ICD-10 codes and it was shared with general practitioners and mental health clinicians to ensure completeness. This resulted in the identification of additional codes which were added.

Individuals with an ICD-10 or READ code from the described list recorded in their GP and/or hospital admission data were identified as having ADHD and/or ASD. The first occurrence of these codes was identified and denoted as age at first available diagnosis. Validation of ADHD codes was also undertaken (see record linkage and validation in online supplemental methods). No nested directly assessed ASD cohort was available to verify the codes used. A full list of the codes used can be seen in online supplemental table S2 while a flow chart of the datasets used and the number of participants at each stage can be seen in online supplemental figures S1 and S2.

The final sample included all individuals born between 1 January 1991 and 31 December 2000 to enable investigation of early adult outcomes during the study period 1 January 2000 to 31 December 2016. For each individual, the follow-up period was defined as their 16th birthday or start of the study period (1 January 2000), whichever came last, to their 25th birthday, end of the study period (31 December 2016), loss to follow-up or death, whichever came first.

### Additional participants
#### Population comparison group
To establish prevalence rates of ADHD and ASD diagnoses in childhood, all individuals within the SAIL Databank (representing the population of Wales) born from 1 January 1991 to 31 December 2000 were identified (n=553551 individuals) and followed to their 18th birthday.

#### Matched-control sample
Three controls for each ADHD/ASD participant were identified. Controls were matched on week of birth (±1 year) and sex, inversely matched by ADHD/ASD diagnosis (ie, controls of cases with ADHD/ASD diagnosed did not have an ADHD/ASD diagnosis), and required to have as much follow-up data as the cases they were matched with. To maximise the quality of ADHD/ASD incidence data in controls used during matching, controls with more data coverage before the age of 18 were selected first.

#### Directly assessed ADHD sample
For comparisons between the electronic cohort and known individuals with ADHD, 579 participants who were given a diagnosis of ADHD using a face-to-face validated diagnostic interview (the Child and Adolescent Psychiatric Assessment)[9] and questionnaire assessments as part of a previous ADHD research study[10] were invited to consent (6–16 years after initial study participation) for their original research data to be anonymously linked to the SAIL Databank. All those contacted had been living in Wales at the time of their assessment. All participants had been originally referred into the study by either a community paediatric or child and adolescent mental health services and had either newly received a diagnosis of ADHD or were suspected as having a diagnosis. Forty-six per cent of participants (n=264) consented to data linkage. Written informed consent was obtained from all young people aged 16 and over, assent from those under 16 and from the parents of those aged 16 years or younger. Participant name, date of birth, address at time of initial research assessment and sex were shared with the NHS Wales informatics service and anonymously linked to the SAIL Databank.

## Outcome measures and covariates

### Anxiety/depression

To assess presence of anxiety and depression disorders, a previously validated common mental health disorders measure[11] was used. This measure used an algorithm to identify all GP or hospital recordings of anxiety or depression disorders and has been shown to have strong positive and negative predictive validity.[11]

### Self-harm

All recorded incidents of self-harm were obtained from GP, emergency department and hospital records. As previously described,[12 13] this included any recorded non-fatal intentional self-harm including self-injury, self-poisoning and suicidal attempts but not suicidal thoughts, regardless of motivation.

### Alcohol use

Alcohol use information was obtained from GP, hospital (inpatient and outpatient) and emergency department datasets, as in John et al.[13] This included recorded incidents of harmful use or dependence syndrome as well as alcohol recorded as being involved in a hospital admission.

### Drug use

As detailed in John et al,[13] drug use information was obtained from GP, hospital (inpatient and outpatient) and emergency department datasets. This consisted of records of any psychoactive substance use diagnoses (except alcohol or tobacco), including harmful use and dependence.

### Emergency department usage

All recorded contacts with hospital emergency departments were obtained from the emergency department dataset.

For anxiety/depression, self-harm, alcohol and drug use measures, records from GP, emergency department and hospital inpatient were combined to obtain an overall measure of any occurrence. ICD-10 and READ codes used for each outcome can be found in their respective cited references.

### Any primary care or hospital use

To investigate whether individuals with ADHD or ASD were more likely to present with the above outcomes at primary care or hospital services (including emergency department use), occurrences of any of the listed outcomes recorded in the General Practice Database (GPD) were considered as primary care use, while any recorded in the Patient Episode Database for Wales (PEDW) or the Emergency Department Dataset (EPD) were considered to be hospital use. These were dichotomised into the presence of any outcome or not. Additional analysis looked at the presence of self-harm separately in GP and hospital records.

### Covariates

Sex and deprivation data were obtained from the Welsh Demographic Service Dataset. Deprivation was assessed using the Welsh Index of Multiple Deprivation (WIMD) obtained for each individual at the start of the follow-up period. The WIMD provides a measure of deprivation by small geographical areas (Lower-layer Super Output Areas) based on seven domains including average income, education, employment and health statistics for individuals in that area. Using the national WIMD cut-offs, quintiles (1–5) of deprivation were defined with level 5 representing the most deprived areas.

The age of each participant at the end of the follow-up period was also recorded to control for the variation in dates of birth and therefore length of follow-up.

## Data analysis

To assess population prevalence, the number of individuals with recorded diagnoses of ADHD and/or ASD was identified within the population comparison sample. This was done separately for males and females.

### Comparison to controls

As the study focused on individuals diagnosed with ADHD or ASD in childhood, individuals with a first recorded diagnosis after the age of 18 years and their matched controls were removed from the dataset.

Cox's regression analyses were undertaken to compare the risk of having a recorded incidence of each outcome for those with ADHD/ASD compared with matched controls, with age at end of follow-up as the time variable and presence/absence of ADHD/ASD as the status variable. Following unadjusted analyses, a second model controlled for sex and proportion of available records and a final model additionally controlled for deprivation.

Cox's regression analyses were also undertaken as sensitivity analyses to investigate risk of early adult outcomes for those with ADHD/ASD and matched controls stratified by sex and separately by deprivation quintile.

Binomial regression analysis was undertaken to investigate whether, in those with recorded self-harm or accident and emergency department usage, the number of incidents of self-harm and emergency department visits was greater in those with ADHD/ASD compared with matched controls. These models were controlled for sex, age at end of follow-up, proportion of available records and deprivation index.

As the SAIL Databank maintains records only from Welsh organisations, individuals who move outside of Wales (and back) or to GPs not covered by the SAIL Databank may not have complete follow-up data. Therefore, analyses were repeated excluding those with <98% total data coverage.

## Public and patient involvement

Understanding the long-term outcomes for children with ADHD was highlighted as a research priority by families at public engagement events. The aims of the study

**Table 1** Overall prevalence of ADHD and ASD, diagnosed in childhood

|  | ADHD* | ASD* | ADHD and ASD | No diagnosis |
|---|---|---|---|---|
| Overall | 7726 (1.4%) | 5000 (0.9%) | 1157 (0.2%) | 541 982 (97.9%) |
| Males† | 6213 (80%) | 3941 (79%) | 978 (85%) | 268 066 (49%) |
| Females† | 1513 (20%) | 1059 (21%) | 179 (15%) | 273 913 (51%) |
| Deprivation‡ |  |  |  |  |
| WIMD quintile: 1§ | 839 (11%) | 756 (15%) | 156 (13%) | 102 867 (20%) |
| WIMD quintile: 2 | 1019 (13%) | 788 (16%) | 163 (14%) | 98 336 (19%) |
| WIMD quintile: 3 | 1325 (17%) | 948 (19%) | 213 (18%) | 93 378 (17%) |
| WIMD quintile: 4 | 1710 (22%) | 1174 (24%) | 260 (23%) | 99 687 (18%) |
| WIMD quintile: 5 | 2833 (37%) | 1329 (27%) | 365 (32%) | 112 861 (21%) |

*Including those with comorbid ADHD and ASD.
†A number of individuals n<5 (all with no diagnosis) had missing data on sex.
‡n=34 858 individuals (n=34 853 with no diagnosis and n=5 with ASD) had missing deprivation data.
§Deprivation quintiles. 5=most deprived areas.
ADHD, attention deficit hyperactivity disorder; ASD, autism spectrum disorder; WIMD, Welsh Index of Multiple Deprivation.

and relevant outcomes were discussed with stakeholders during the study design stage.

## RESULTS

For a flow chart of the databases used and the number of participants used at each stage, see online supplemental figure S2. Average coverage of the datasets was comparable with other cohorts (eg, John *et al*[13]) with 99.4% coverage from PEDW and 65.2% average coverage of GPD of the time prior to 18 years.

### Identification of ADHD and ASD diagnoses and prevalence

Within the e-cohort, a total of 7726 individuals had a recorded diagnosis of ADHD and 5001 of ASD by the age of 18 years (see table 1). This represents 1.4% and 0.9% of the total Welsh population within that age group. 0.2% (n=1157) had both ADHD and ASD with 22.5% of all individuals with ADHD having a comorbid diagnosis of ASD and 15.3% of all individuals with ASD having a comorbid diagnosis of ADHD. Males were more likely than females to have a diagnosis of ADHD with 2.2% of males and 0.6% of females receiving such a diagnosis. The same pattern was observed for those with ASD with 1.5% of males and 0.4% of females receiving such a diagnosis.

### Comparison between those with ADHD and matched controls

A total of 19 532 individuals were identified as matched controls for ADHD individuals. While matched on age and sex, those with ADHD came from slightly more deprived backgrounds than their matched controls (see table 2). The mean age at diagnosis for those with ADHD was 9.87 years (SD: 3.81). A summary of the associations between ADHD and outcomes can be seen in figure 1.

Controlling for age, those with ADHD had approximately a twofold increase in risk of having a record of anxiety or depression (HR: 2.36, 95% CI: 2.20 to 2.53) and a fivefold increase in risk of self-harm (HR: 5.70,

95% CI: 5.07 to 6.40). They were significantly more likely to have a record of alcohol (HR: 3.95, 95% CI: 3.42 to 4.56) and drug use (HR: 5.88, 95% CI: 5.08 to 6.80) and were more likely to have used emergency department services (HR: 1.36, 95% CI: 1.31 to 1.41). All associations remained when controlling for sex, record availability and deprivation (online supplemental table S3). For those with recorded self-harm or emergency department visits, those with ADHD had significantly more frequent visits than controls (online supplemental table S4). Increased risk of self-harm was observed using both GP (HR: 6.5, 95% CI: 5.65 to 7.72) and hospital records (HR: 7.07, 95% CI: 5.91 to 8.48) (online supplemental table S5). When stratifying the outcomes by sex, both males and females were at higher risk for all outcomes and there was generally no difference in the magnitude of this risk, except that females with ADHD were more likely to use emergency department and hospital services than males with ADHD (figure 2 and online supplemental tables S6 and S7). Similarly, there were no clear differences in the magnitude of risks by level of deprivation (online supplemental figures S3 and S8). When those with incomplete data coverage were removed from the analyses (n=1696, 6.4%), there were no changes in the findings (online supplemental table S9).

### Comparison between those with ASD and matched controls

A total of 11 427 individuals were identified as matched controls for the 5001 individuals with ASD. Those with ASD came from similar deprivation backgrounds as their matched controls (see table 2). The mean age of diagnosis for ASD was 10.02 years (SD: 4.52). A summary of the difference between the prevalence of the early adult outcomes between these two groups can be seen in figure 1. Those with ASD had approximately a twofold risk of having a record of anxiety/depression (HR: 2.11, 95% CI: 1.91 to 2.34) and were almost three times more likely

**Table 2** Summary of demographics and adult outcomes for ADHD and ASD cases and controls

| | ADHD controls | ADHD cases | ASD controls | ASD cases |
|---|---|---|---|---|
| N | 19 532 | 7738 | 11 427 | 5001 |
| Male gender | 15 568 (79.7%) | 6223 (80.4%) | 8817 (77.2%) | 3945 (78.9%) |
| Age at end of follow-up (GP records): mean (SD) | 20.4 (2.8) | 19.97 (2.7) | 19.6 (2.8) | 19.4 (2.6) |
| Deprivation index quintile (start of follow-up): mean (SD) | 3.2 (1.5) | 3.4 (1.4) | 3.1 (1.4) | 3.2 (1.4) |
| Age at ADHD diagnosis: mean (SD) | – | 9.8 (3.8) | – | 8.9 (4.2) |
| Age at ASD diagnosis: mean (SD) | – | 10.3 (4.9) | – | 10.0 (4.5) |
| Anxiety or depression: N (%) | 1879 (9.6) | 1324 (17.1) | 919 (8.0) | 653 (13.1) |
| Drug use—any: N (%) | 284 (1.5) | 497 (6.4) | 110 (1.0) | 80 (1.6) |
| Alcohol use—any: N (%) | 349 (1.8) | 408 (5.3) | 192 (1.7) | 76 (1.5) |
| Self-harm—any: N (%) | 447 (2.3) | 769 (9.9) | 243 (2.1) | 241 (4.8) |
| Self-harm—number of recorded incidents: mean (SD) | 0.04 (0.37) | 0.28 (1.41) | 0.04 (0.37) | 0.16 (1.29) |
| Emergency department use—any: N (%) | 10 550 (54) | 4899 (62) | 5668 (50) | 2115 (42.3) |
| Emergency department use—number of recorded visits: mean (SD) | 1.67 (2.61) | 3.07 (6.19) | 1.45 (2.57) | 1.54 (3.95) |

ADHD, attention deficit hyperactivity disorder; ASD, autism spectrum disorder; GP, general practice.

to have a record of self-harm (HR: 2.93, 95% CI: 2.45 to 3.50). They were more likely to have a record of drug use (HR: 2.21, 95% CI: 1.66 to 2.95) but showed similar levels of alcohol use problems (HR: 1.19, 95% CI: 0.91 to 1.55) and rates of emergency department use (HR: 0.95, 95% CI: 0.90 to 1.00) (see figure 1 and table 2). Associations

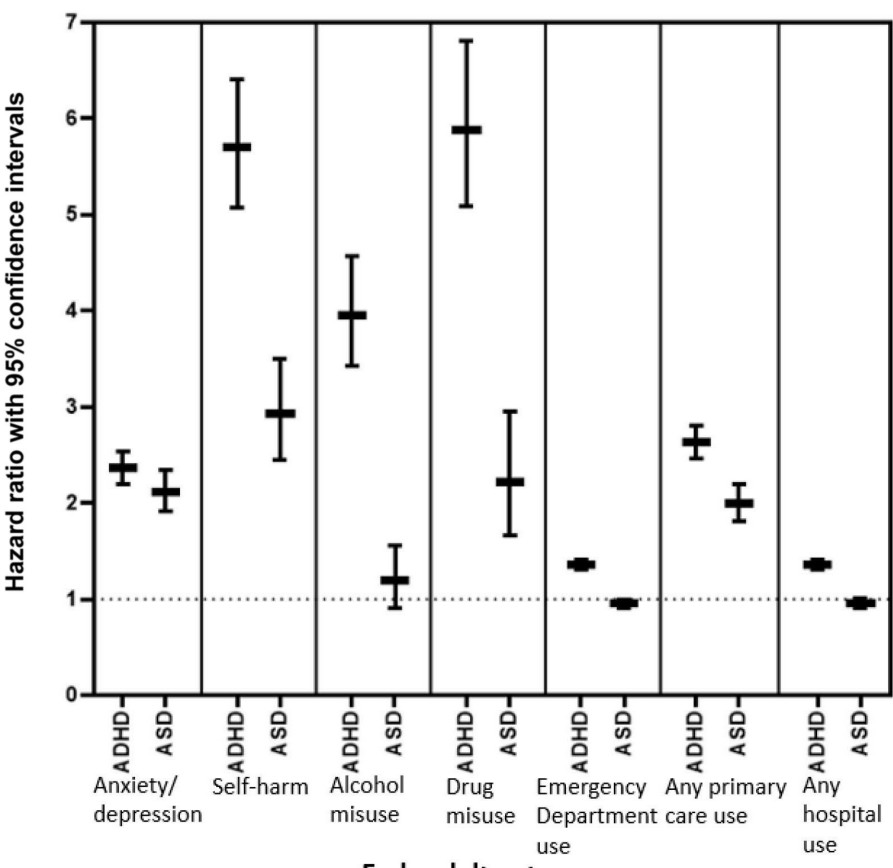

**Figure 1** Associations of ADHD and ASD with early adult outcomes. ADHD, attention deficit hyperactivity disorder; ASD, autism spectrum disorder.

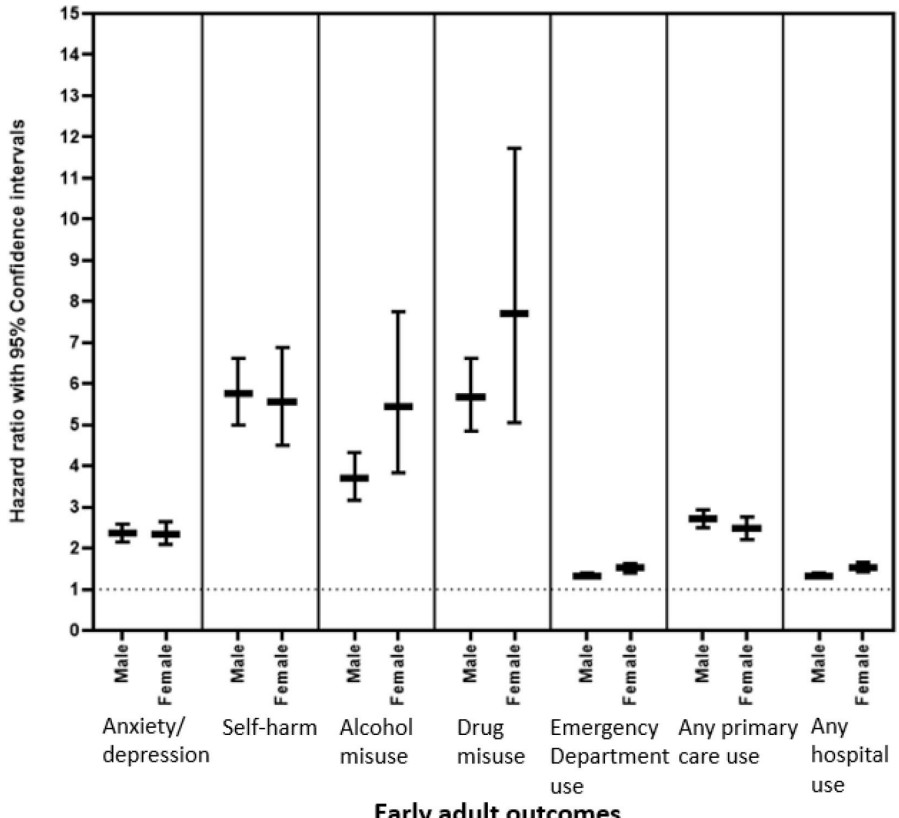

**Figure 2** Associations between ADHD and early adult outcomes, by sex. ADHD, attention deficit hyperactivity disorder.

remained robust controlling for sex, record availability and deprivation (see online supplemental table S10). An ASD diagnosis was associated with slightly reduced self-harm; however for in those who self-harmed, those with ASD were more likely to have repeated self-harm than controls. Increased risk of self-harm was observed in GP (HR: 3.60, 95% CI: 2.86 to 4.53) and hospital records (HR: 3.75, 95% CI: 2.85 to 4.93) (see online supplemental table S11). For those who used emergency department care, those with ASD had significantly more visits than their matched controls (online supplemental table S4). As seen in figure 3 and online supplemental tables 12 and 13, findings were similar for males and females and, in general there were no differences in the risks across sex. The exception to this was that females with ASD were more likely to use emergency department and hospital services than males with ASD. When stratified by deprivation (online supplemental table S14 and figure S4), there was no clear differences in the findings by deprivation. When those with incomplete coverage were removed from the analyses (n=1004, 6.4%), the findings remained the same (online supplemental table S15).

### Comparison of directly assessed ADHD sample to those identified in the e-cohort

Within the directly assessed sample, 154 individuals who were born between 1991 and 2000 and were included in the matched sample dataset. The directly assessed ADHD cohort were younger at the end of follow-up than those in the e-cohort only (mean: 19.11 years, SD: 2.20 vs mean:

21.20 years, SD: 3.80; t=11.320, p<0.001) and had an earlier age at ADHD diagnosis (mean: 8.41 years, SD: 3.34 vs mean: 10.54 years, SD: 3 .86; t=7.949, p<0.001). There groups were similar on other demographic measures (online supplemental table S16). The difference in age at diagnosis remained when controlling for age, sex, proportion of follow-up records and deprivation (unst. beta:−1.342, 95% CI:−1.91 to −0.774, beta=−0.045). When controlling for length of follow-up, there were no significant differences between those in the hybrid and e-cohort for all outcome measures.

### DISCUSSION

The aims of this study were to establish and validate a nationwide e-cohort of young people in Wales with recorded diagnoses of ADHD and/or ASD. These aims were successfully achieved; lists of relevant ICD-10 and NHS READ codes were created and used to identify an e-cohort of individuals with ADHD and/or ASD. In comparisons between these groups and matched population controls, we observed associations of ADHD or ASD with adverse early adult outcomes in line with research findings from the UK and other countries, validating the identification strategy.[2 3 14–17] Further validating the process, those with ADHD in the e-cohort were compared with a hybrid cohort of individuals with known ADHD diagnoses. These groups did not differ in their early adult outcomes and were demographically similar, apart

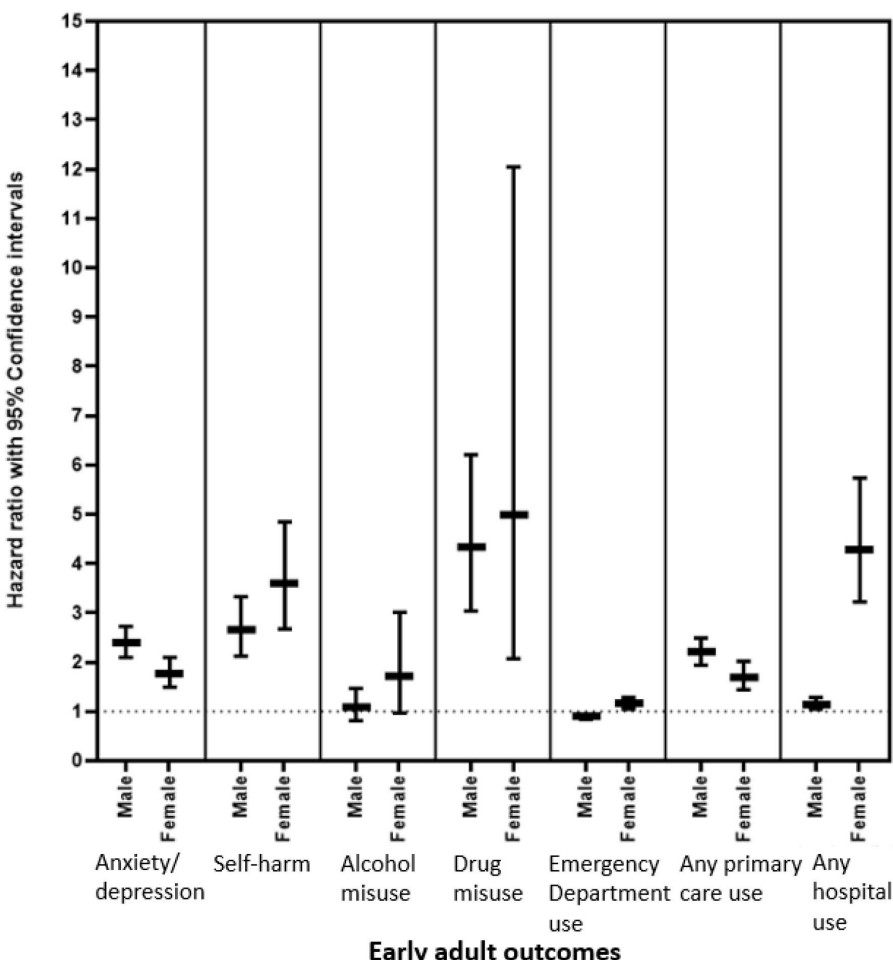

**Figure 3** Associations between ASD and early adult outcomes, by sex. ASD, autism spectrum disorder.

from that those in the hybrid cohort were younger and had an earlier age at diagnosis. The identification of diagnosed individuals used data from multiple databases (GP, hospital and prescription records) which reinforces presumed assumptions that there is no single code or database routinely used for recording for ADHD or ASD diagnoses within Wales (and the UK). This is in contrast to the national registries in some other countries, such as Sweden and Denmark where diagnoses are routinely recorded using ICD and standardised prescription codes (eg, Wimberley et al[18] and Chang et al[19]). In the UK (as in some other countries), diagnoses of ADHD and ASD are made by specialists and not by primary care but ICD codes are not always used.

The administrative prevalence of individuals with a recorded diagnosis of ADHD and ASD (1.4% and 0.9% of the respectively) in Wales, identified in the SAIL Databank corresponds well with the population prevalence of these disorders from the largest, recent UK epidemiological survey: the 2017 Mental Health of Children and Young People survey[20] which directly assessed a representative sample of 5–19 year olds in England (not Wales) using the Development and Well-Being Assessment, a structured diagnostic interview.[21] Rates of DSM-5 diagnoses reported in that study were 1.6% for ADHD and 1.2% for ASD.

The rate of ADHD in that survey is much lower than the rates reported in meta-analyses of ADHD which suggest global prevalence rates of around 5%–7% with variation explained by methodological differences across study rather than because of substantial differences between countries.[22] This may explain why an earlier English mental health population survey[23] observed a prevalence rate of 2.4% for DSM-IV-defined ADHD. Population prevalence rates for ADHD symptom levels do not appear to have changed over time.[22 24] Administrative rates of ADHD have varied considerably across time and different countries with the UK and Scandinavian countries generally showing a rate commensurate with or lower than the population prevalence rate[25 26] while some American States show very high administrative prevalence rates.[27]

In contrast, population rates of autism have increased over time.[28] This is thought to be mainly attributable to changes in diagnostic criteria and methods of assessment.[29] However, the population rate of autism in 2005 in England (0.9%) is[30] similar to that reported in the most recent 2017 survey.[20] The prevalence rates for males and females with an excess of males affected in our study are in line with population and administrative surveys.[31 32] A recent study looking at rates of autism also using the SAIL Databank, using similar methodology, found a

slightly lower prevalence rate than in this study (0.5%).[33] However, this difference is likely to be due to the age range of the sample and the known increase in diagnostic prevalence over time as that study looked at individuals with autism at any age, while we restricted to a diagnosis before the age of 18 years.

Our study investigated diagnoses recorded in healthcare. If we use findings from the 2017 England population survey, this suggests that there is neither an overdiagnosis or underdiagnosis of ADHD or ASD in Wales. However, using data from other surveys would suggest that there could be underdiagnosis of ADHD but that administrative ASD rates are in line with nearly all population surveys. In this study, we also identified those with both ADHD and ASD despite codiagnosis being disallowed by ICD and DSM until recently; these are findings that are also observed in the Danish registry (eg, Ottosen *et al*[34]). This suggests that clinicians, unlike researchers, are more willing to dispense with diagnostic rules. However, it is possible that future estimates of ADHD and ASD codiagnosis will be higher since DSM-5 and ICD-11 no longer disallow this.[35] Looking at adverse outcomes in early adulthood, as anticipated, ADHD was significantly associated with all the outcomes studied. Perhaps most strikingly, there was more than a fivefold increased risk of self-harm in individuals in this group, while we also observed that, of those with recorded self-harm events, those with ADHD had an increased number of incidents. While this increased risk of self-harm in those with ADHD has previously been reported (eg, Alley[36] especially when looking specifically at suicidal behaviour), our study extends previous studies as it includes self-harm behaviours recorded in primary as well as secondary care.

A recorded diagnosis of ASD was associated with increased risk of anxiety/depression, self-harm and drug use. Again, these are in line with previous literature,[17 37 38] thereby validating findings from this e-cohort. Interestingly, those with ASD showed similar levels of alcohol use problems and emergency department use for those with ASD, although of those who did have contact with emergency departments, had more visits than their matched controls. While an increased risk was observed for those with ADHD or ASD across a range of adverse outcomes, this risk tended to be greater for those with ADHD for all outcomes except anxiety/depression where there was an equally increased risk.

Sensitivity analyses suggest that adverse outcomes for ADHD and ASD are similar in males and females and across economic/deprivation strata. Comparisons of the sample of individuals with a diagnosis of ADHD confirmed by research diagnostic interviews who consented to record linkage and the generated e-cohort demonstrated that they were generally similar, although individuals within the hybrid cohort were younger at the end of the follow-up period and were also likely to be younger at age of diagnosis. As participants were part of a larger research study who were recontacted to request consent for data linkage, the reduced follow-up period, indicating a younger hybrid cohort, may reflect the difficulties of continued contact with research participants as they move into adulthood. This highlights the difficulties of longitudinal follow-up studies, further demonstrating the benefits of linking routinely collected health data for assessing long-term outcomes. When the length of follow-up was taken into consideration, individuals with ADHD in the hybrid and e-cohort were similar, indicating that the e-cohort successfully identified individuals reflective of those with ADHD in the Welsh population.

This study demonstrates that health record data for a whole nation via the SAIL Databank can be used to identify those with a diagnosis of ADHD and ASD and enabling (anonymous) longitudinal follow-up for all individuals still living in Wales, reducing issues associated with selective attrition that can be a disadvantage for longitudinal cohort studies.[5–7] By looking at the population level, we have a large sample of individuals with ADHD/ASD and so can look at a range of outcomes including those which are less common, such as drug and alcohol use. Our study also highlights the opportunity for other types of large-scale studies including examining early risk factors, service use, physical health, educational and social outcomes (via linkage with education and social care records in Wales).

The verification of ADHD codes using a directly assessed sample as comparison was successful and gives confidence that the e-cohort is similar to those with a known diagnosis. However, there are limitations. This verification process could have been improved by including a directly assessed sample without ADHD to assess the false positive rate; however, this was not available. Similarly, verification of the ASD codes would have been advantageous. However, identifying a sample of individuals of similar prevalence to the most recent population survey in England[20] and the observed association of ASD with adverse early adult outcomes increases confidence in our identification methods.

While using a good set of verified codes, as with any data linkage study, we were reliant on the information recorded in the relevant dataset. Therefore, individuals whose ADHD/ASD has not been assessed or diagnosed clinically or where such diagnoses were not accurately recorded would not be included in our sample. This may also be the case for the outcome measures; although we used previously established methods to identify these, if they are not recorded (eg, self-harm not reported to healthcare services, and the known underestimation of community alcohol and drug use) they would not be represented in our sample. It is also the case that the follow-up period differed between individuals and was not long for the youngest in our cohort, thus potentially underestimating the prevalence of the outcomes. It is worth noting that these limitations are relevant to the matched controls as well as the individuals with recorded ADHD/ASD diagnoses and that the unique availability of records from both primary and secondary care in the SAIL Databank may mitigate reporting issues.

Another limitation is that we also restricted the sample to those with a diagnosis of ADHD/ASD by age 18 years, as we were interested in what happens to those with a childhood diagnosis as they grow older. However, we know that many individuals are diagnosed with ADHD and ASD in adulthood (especially females) and investigating their difficulties, especially in comparison to those with a childhood diagnosis, would be interesting.

Overall, this study has demonstrated the feasibility and validity of an e-cohort of those with ADHD and ASD using health record-linked data in Wales, UK. Our identification strategy resulted in a large sample who could be followed longitudinally into early adulthood with minimal attrition. This type of cohort can be used in future studies, benefitting from the linkage of information from primary and secondary healthcare, social care and educational data available via the SAIL Databank, to further understand the risks and outcomes for those with ADHD and ASD as well as to support planning in health and social care services.

**Author affiliations**
[1]School of Psychology, Cardiff University, Cardiff, UK
[2]Population Data Science, Swansea University, Swansea, UK
[3]Wolfson Centre for Young People's Mental Health, Cardiff University, Cardiff, UK
[4]National Centre for Register-based Research, School of Business and Social Sciences, Aarhus University, Aarhus, Denmark
[5]Institute of Clinical Medicine, University of Copenhagen, Copenhagen, Denmark
[6]Department of Child and Adolescent Psychiatry, Mental Health Services of the Capital Region, Glostrup, Denmark
[7]Medical Sciences and Nutrition, University of Aberdeen, Aberdeen, UK
[8]Division of Psychological Medicine and Clinical Neurosciences; Centre for Neuropsychiatric Genetics and Genomics, School of Medicine, Cardiff University, Cardiff, UK

**Acknowledgements** We thank Val Russell and Jan Whitley for their administrative support on this project. We thank the clinicians who supported the identification of relevant administrative codes, especially Ajay Thapar and Rob Potter. We also thank all of the families from the SAGE study who contributed to the directly assessed cohort and helped develop the aims and outcomes included in this study through public engagement events.

**Contributors** Concept and design: KL, AJ, SP and AT. Acquisition, analysis or interpretation of data: KL, MDP-B, SD, SP, LR, AJ and AT. Drafting of the manuscript: KL, MDP-B and AT. Statistical analysis: KL and MDP-B. Critical revision of the manuscript for important intellectual content: KL, MDP-B, SD, SP, LR, AJ and AT. KL and MDP-B had access to all the data in the study and take responsibility for the integrity of the data and the accuracy of the data analysis. KL is the guarantor.

**Funding** This study was funded by a Wellcome Trust Institutional Strategic Support Fund awarded by Cardiff University, Grant number (Grant ref: 105613/Z/14/Z) and the National Centre for Mental Health (NCMH) which is funded by Health and Care Research Wales (Grant ref: 517191). LR is supported by the Wolfson Foundation. As this research was funding in whole, or in part, by the Wellcome Trust, for the purpose of open access, the author has applied a CC BY public copyright license to any Author Accepted Manuscript version arising from this submission.

**Competing interests** AT is on the executive board of the UK Charity ADHD Foundation (unpaid). All salary comes from Cardiff University only. KL has received a speaker's fee from Medice on a topic unrelated to this research. All other authors declare no conflicts of interest.

**Patient and public involvement** Patients and/or the public were involved in the design, or conduct, or reporting, or dissemination plans of this research. Refer to the Methods section for further details.

**Patient consent for publication** Not applicable.

**Ethics approval** This study was approved by the NHS Wales Research Ethics Committee (14/WA/0157). Access to data from the SAIL Databank was approved by the Information Governance Review Panel (IGRP) (project number 0712). Participants gave informed consent to participate in the study before taking part.

**Provenance and peer review** Not commissioned; externally peer reviewed.

**Data availability statement** Data may be obtained from a third party and are not publicly available. Data within this study are available in the SAIL Databank. Access to the SAIL Databank is subject to review by and independent Information Governance Review Panel (IGRP). The IGRP gives careful consideration to each project to ensure proper and appropriate use of SAIL data, and approval from the IGRP is necessary to access any data. Once access has been granted, it is gained through the SAIL Gateway: a privacy protecting safe haven and remote access system. Details on how to access data from the SAIL Databank are available here: https://saildatabank.com/data/.

**ORCID iDs**
Kate Langley http://orcid.org/0000-0002-2033-2657
Shantini Paranjothy http://orcid.org/0000-0002-0528-3121
Ann John http://orcid.org/0000-0002-5657-6995
Anita Thapar http://orcid.org/0000-0002-3689-737X

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
