## [Reviewer comments · BMJ Open]

ARTICLE DETAILS

TITLE (PROVISIONAL)	Can a nation-wide e-cohort of ADHD and ASD in Childhood be established using Welsh routinely available datasets?
AUTHORS	Langley, Kate; Del Pozo-Banos, Marcos; Daalsgard, Søren; Paranjothy, Shantini; Riglin, Lucy; John, Ann; Thapar, Anita

VERSION 1 – REVIEW

REVIEWER	Cybulski, Lukasz King's College London Institute of Psychiatry Psychology and Neuroscience
REVIEW RETURNED	19-Mar-2023

GENERAL COMMENTS	Please find my comments below: 1. A stated aim of the study was to establish the feasibility of identifying individuals with a diagnosis of ADHD and ASD using routinely collected clinical data. However, primary care (e.g., Clinical Practice Research Datalink, QResearch) and hospital records (e.g., Hospital Episode Statistics) have been used in the UK for such investigations for over two decades. With this in mind, I would think that the feasibility of using routinely collected primary care and hospital data for research purposes already has been demonstrated. If not, then the authors should perhaps explain this more clearly in the introduction and how the present investigation addresses this issue.2. The authors rightly identify the use of unvalidated administrative codes (i.e., Read codes) as a potential threat to the validity of investigations based on electronic health care records, but it is only really mentioned in the discussion. Since a stated aim of the study is validation, it would make sense to discuss it in the introduction at greater length, and make it clear how the study addresses this issue.3. I also wonder to what extent the authors have been successful in their efforts to validate diagnoses. Firstly, the comparison between cases identified in the e-cohort and the research study was limited to ADHD. Second, the nature of this validation was somewhat limited in that it couldn't establish a false-positive rate among cases identified in the e-cohort. The comparison with respect to outcomes and demographic characteristics is indirect and using it as a basis for concluding validity seems unduly optimistic. I appreciate that the authors are aware of this, and rightly identify this is a limitation, but with this in mind I wonder if the study has actually gone over and beyond what is necessary to reasonably establish validity.
---

	4. The rationale for and discussion around the choice of outcomes could be improved. For instance, self-harm outcomes have been examined extensively in ADHD and ASD cohorts delineated using routinely collected clinical records: Chen, M. H., Pan, T. L., Lan, W. H., Hsu, J. W., Huang, K. L., Su, T. P., ... & Bai, Y. M. (2017). Risk of suicide attempts among adolescents and young adults with autism spectrum disorder: A nationwide longitudinal follow-up study. The Journal of clinical psychiatry, 78(9), 1709. Chou, I. C., Lin, C. C., Sung, F. C., & Kao, C. H. (2014). Attention-deficit hyperactivity disorder increases the risk of deliberate self-poisoning: A population-based cohort. European psychiatry, 29(8), 523-527. Cybulski, L., Ashcroft, D. M., Carr, M. J., Garg, S., Chew-Graham, C. A., Kapur, N., & Webb, R. T. (2022). Risk factors for nonfatal self-harm and suicide among adolescents: two nested case-control studies conducted in the UK Clinical Practice Research Datalink. Journal of child psychology and psychiatry, 63(9), 1078-1088 Huang, K. L., Wei, H. T., Hsu, J. W., Bai, Y. M., Su, T. P., Li, C. T., ... & Chen, M. H. (2018). Risk of suicide attempts in adolescents and young adults with attention-deficit hyperactivity disorder: a nationwide longitudinal study. The British Journal of Psychiatry, 212(4), 234-238. Thus, the idea advanced in the introduction that adverse outcomes have only been examined in smaller prospective cohort studies with substantial attrition isn't necessarily true. I am not suggesting that the authors must cite these specific papers, but they ought to acknowledge the existing literature on self-harm and all the other outcomes as well. For example, have depression/anxiety/alcohol and drug use been examined before as endpoints in ADHD/ASD cohorts delineated using routinely collected clinical data? If they have not been examined, then it should be investigated and discussed in the introduction and perhaps used as a rationale for the present study. At the moment it is isn't very clear what the study is adding to the broader literature in this respect. 5. This relates to the previous point (4): long-term outcomes were identified as a priority for families during public engagement activities. In my view this is not sufficient a rationale if previous studies have already examined these outcomes. I think you need to establish a research rationale for investigating these outcomes as well.
--	--

REVIEWER	Canals, Josefa Universitat Rovira i Virgili, Psicologia
REVIEW RETURNED	08-May-2023

GENERAL COMMENTS	The manuscript addresses an under-evaluated but interesting objective in the field of public health and especially in the study of mental health disorders, in this case ADHD and ASD. It is well written and the study design and the method are appropriate to answer the objectives. The
---

	results are presented clearly through the text, tables, and much supplementary material (Figures and Tables) This study demonstrates that health record data for a whole nation via the SAIL Databank can be used to identify those with a diagnosis of ADHD and ASD and enabling (anonymous) longitudinal follow up for all individuals still living in Wales. This makes the authors handle a large sample of individuals with ADHD/ASD and of recorded data of the population, including matched controls. The authors take into account the different limitations of the study, which I share. However, other aspects could be added, if they cannot be answered (see below). Recommendations to the authors:  - Data of the sample were from subjects with a diagnosis of ADHD/ASD by age 18 years. The authors report mean age of the diagnoses which is quite late for a diagnosis of ASD. Although the research question is to relate in general the presence of anxiety/depression, drug/alcohol use, self-harm and Emergency Department use, it would be very interesting to know if early or late diagnosis (even separating groups according to age of diagnosis) is a predictor variable for the recorded outcomes or for others outcomes in future. Prior to diagnosis, did these children have academic dysfunction or other clinical health problems? - It would also be important to know data on contacts with social or legal services, and school data. - Add significance level values in the results, mainly in the Tables. In this sense, there are significant differences between quintiles according to diagnosis?. Also include these values in the statistical results in the supplementary tables.
--	--

VERSION 1 – AUTHOR RESPONSE

Reviewer 1:

*A stated aim of the study was to establish the feasibility of identifying individuals with a diagnosis of ADHD and ASD using routinely collected clinical data. However, primary care (e.g., Clinical Practice Research Datalink, QResearch) and hospital records (e.g., Hospital Episode Statistics) have been used in the UK for such investigations for over two decades. With this in mind, I would think that the feasibility of using routinely collected primary care and hospital data for research purposes already has been demonstrated. If not, then the authors should perhaps explain this more clearly in the introduction and how the present investigation addresses this issue.

We agree that health-record data has been used previously this study provides for ADHD or ASD across the whole of an entire UK nation. The aim of this particular study was to establish the ADHD/ASD cohort within the established dataset of routinely collected clinical data. It represents the Welsh context, a devolved nation with its own health policy, pattern of care and which is not well represented in the CPRD or QResearch databases. The SAIL databank also has a much more comprehensive coverage of the whole population than these other datasets (77% vs. 13% for CPRD (Wolf et al., 2019, <https://doi.org/10.1093/ije/dyz034>) for primary care data linked to hospital episodes and area deprivation) The SAIL databank has the additional advantages over the mentioned datasets

such as having whole population linked emergency department attendance data. This has now been specified further in the introduction (page 5).

2. The authors rightly identify the use of unvalidated administrative codes (i.e., Read codes) as a potential threat to the validity of investigations based on electronic health care records, but it is only really mentioned in the discussion. Since a stated aim of the study is validation, it would make sense to discuss it in the introduction at greater length, and make it clear how the study addresses this issue.

We have added this information to the introduction (page 5).

3. I also wonder to what extent the authors have been successful in their efforts to validate diagnoses. Firstly, the comparison between cases identified in the e-cohort and the research study was limited to ADHD. Second, the nature of this validation was somewhat limited in that it couldn't establish a false-positive rate among cases identified in the e-cohort. The comparison with respect to outcomes and demographic characteristics is indirect and using it as a basis for concluding validity seems unduly optimistic. I appreciate that the authors are aware of this, and rightly identify this is a limitation, but with this in mind I wonder if the study has actually gone over and beyond what is necessary to reasonably establish validity.

As noted, we agree with the reviewer about the limitations of our approach and as noted have highlighted these within the discussion section (which has been briefly added to, page 19). As samples were not available to undertake these additional verification steps, we hope that the readers will appreciate that we have been up front about these limitations in the discussion section.

4. The rationale for and discussion around the choice of outcomes could be improved. For instance, self-harm outcomes have been examined extensively in ADHD and ASD cohorts delineated using routinely collected clinical records:

Chen, M. H., Pan, T. L., Lan, W. H., Hsu, J. W., Huang, K. L., Su, T. P., ... & Bai, Y. M. (2017). Risk of suicide attempts among adolescents and young adults with autism spectrum disorder: A nationwide longitudinal follow-up study. *The Journal of clinical psychiatry*, 78(9), 1709.

Chou, I. C., Lin, C. C., Sung, F. C., & Kao, C. H. (2014). Attention-deficit hyperactivity disorder increases the risk of deliberate self-poisoning: A population-based cohort. *European psychiatry*, 29(8), 523-527.

Cybulski, L., Ashcroft, D. M., Carr, M. J., Garg, S., Chew-Graham, C. A., Kapur, N., & Webb, R. T. (2022). Risk factors for nonfatal self-harm and suicide among adolescents: two nested case-control studies conducted in the UK Clinical Practice Research Datalink. *Journal of child psychology and psychiatry*, 63(9), 1078-1088

Huang, K. L., Wei, H. T., Hsu, J. W., Bai, Y. M., Su, T. P., Li, C. T., ... & Chen, M. H. (2018). Risk of suicide attempts in adolescents and young adults with attention-deficit hyperactivity disorder: a nationwide longitudinal study. *The British Journal of Psychiatry*, 212(4), 234-238.

Thus, the idea advanced in the introduction that adverse outcomes have only been examined in smaller prospective cohort studies with substantial attrition isn't necessarily true. I am not suggesting that the authors must cite these specific papers, but they ought to acknowledge the existing literature on self-harm and all the other outcomes as well. For example, have depression/anxiety/alcohol and drug use been examined before as endpoints in ADHD/ASD cohorts delineated using routinely collected clinical data? If they have not been examined, then it should be investigated and discussed in the introduction and perhaps used as a rationale for the present study. At the moment it is isn't very clear what the study is adding to the broader literature in this respect.

As the reviewer suggests, our aim had been to utilise known adverse outcomes for those with ADHD and ASD to examine the validity of our cohort whilst this also provides some specific Welsh data. Our intention in the introduction was not to suggest that there are no other studies using routine healthcare data, rather to point out the advantages of this method beyond other cohort designs. We

have now explicitly mentioned this in the introduction (page 5). We have also highlighted that these findings were not unexpected within the discussion section (page 18). We would argue that our findings around e.g. self-harm are still worthy of discussion, as the combination of both primary and secondary care records, the comprehensive use of both READ codes and (for ADHD) prescription data and the age of our sample, as well as looking across a range of types of self-harm is not considered in previous studies including those that the reviewer has cited.

5. This relates to the previous point (4): long-term outcomes were identified as a priority for families during public engagement activities. In my view this is not sufficient a rationale if previous studies have already examined these outcomes. I think you need to establish a research rationale for investigating these outcomes as well.

We wholeheartedly agree that this study does not meaningfully address the identified priorities of families in itself. The establishment of a relevant cohort was identified as a first step to address these priorities. This has now been made clear (page 8).

Reviewer 2:

Recommendations to the authors:

- Data of the sample were from subjects with a diagnosis of ADHD/ASD by age 18 years. The authors report mean age of the diagnoses which is quite late for a diagnosis of ASD. Although the research question is to relate in general the presence of anxiety/depression, drug/alcohol use, self-harm and Emergency Department use, it would be very interesting to know if early or late diagnosis (even separating groups according to age of diagnosis) is a predictor variable for the recorded outcomes or for others outcomes in future. Prior to diagnosis, did these children have academic dysfunction or other clinical health problems?

We agree that this would be an interesting additional question. As noted in the manuscript (page x) we report the age at first recorded diagnosis. As there is not an official, systematic way of recording diagnoses of ADHD or ASD within READ codes or via the use of prescribed medication, it is possible that the actual date of diagnosis was earlier than the recorded date. We therefore decided not to do comparisons between those diagnosed early and late or to look at service use prior to recorded diagnosis date.

- It would also be important to know data on contacts with social or legal services, and school data.

We agree that these would be interesting additional outcomes to investigate but were beyond the scope of the current funded investigation and were not included in our current dataset. We aim to look at this in future studies.

- Add significance level values in the results, mainly in the Tables. In this sense, there are significant differences between quintiles according to diagnosis?. Also include these values in the statistical results in the supplementary tables

We have not presented significance levels in our results due to reported issues of overreliance on arbitrary significant differences and other reported issues (see e.g. Sterne JAC, Davey Smith G. Sifting the evidence—what's wrong with significance tests? British Medical Journal 2001: 322; 226-231). Instead, we report the 95% confidence intervals for our analyses to allow interpretation of relevance.

VERSION 2 – REVIEW

REVIEWER	Cybulski, Lukasz King's College London Institute of Psychiatry Psychology and Neuroscience
REVIEW RETURNED	20-Jul-2023

GENERAL COMMENTS

I'm happy with the changes made by the authors.

VERSION 2 – AUTHOR RESPONSE